# Epigenetic Modulations for Prevention of Infectious Diseases in Shrimp Aquaculture

**DOI:** 10.3390/genes14091682

**Published:** 2023-08-25

**Authors:** Gunasekara Chathura Wikumpriya, Madhuranga Walawedurage Srinith Prabhatha, Jiye Lee, Chan-Hee Kim

**Affiliations:** Division of Fisheries Life Science, Pukyong National University, Busan 48513, Republic of Koreasrinith123@pukyong.ac.kr (M.W.S.P.); jiyelee@pukyong.ac.kr (J.L.)

**Keywords:** shrimps, disease management, epigenetics, epigenetic modulation, future applications

## Abstract

Aquaculture assumes a pivotal role in meeting the escalating global food demand, and shrimp farming, in particular, holds a significant role in the global economy and food security, providing a rich source of nutrients for human consumption. Nonetheless, the industry faces formidable challenges, primarily attributed to disease outbreaks and the diminishing efficacy of conventional disease management approaches, such as antibiotic usage. Consequently, there is an urgent imperative to explore alternative strategies to ensure the sustainability of the industry. In this context, the field of epigenetics emerges as a promising avenue for combating infectious diseases in shrimp aquaculture. Epigenetic modulations entail chemical alterations in DNA and proteins, orchestrating gene expression patterns without modifying the underlying DNA sequence through DNA methylation, histone modifications, and non-coding RNA molecules. Utilizing epigenetic mechanisms presents an opportunity to enhance immune gene expression and bolster disease resistance in shrimp, thereby contributing to disease management strategies and optimizing shrimp health and productivity. Additionally, the concept of epigenetic inheritability in marine animals holds immense potential for the future of the shrimp farming industry. To this end, this comprehensive review thoroughly explores the dynamics of epigenetic modulations in shrimp aquaculture, with a particular emphasis on its pivotal role in disease management. It conveys the significance of harnessing advantageous epigenetic changes to ensure the long-term viability of shrimp farming while deliberating on the potential consequences of these interventions. Overall, this appraisal highlights the promising trajectory of epigenetic applications, propelling the field toward strengthening sustainability in shrimp aquaculture.

## 1. Introduction

In the face of a rapidly expanding global population, ensuring food security has become an increasingly formidable challenge. To address this challenge, the aquaculture sector plays an important role in meeting the escalating worldwide demand for food. In 2020, the aquaculture industry achieved a momentous feat, attaining a record-breaking aquatic production of 122.6 million tonnes (valued at USD 281.5 billion), and it has expanded to 184.6 million tonnes in 2022 [1]. Global aquaculture primarily centers on the commercial cultivation of a diverse range of marine vertebrates, such as teleost fish, and invertebrates, including mollusks and crustaceans [2]. Among teleost fish, the prevailing species cultivated in the aquaculture sector encompass carp, catfish, salmon, and tilapia, while shrimp and crabs are the two most invaluable crustacean commodities traded on a global scale [3,4].

Shrimp is a highly prized seafood replete with nutrients like protein, omega-3 fatty acids, and vitamins, and this industry plays a critical role in the global economic landscape and food security [5,6]. According to the Food and Agriculture Organization (FAO), farmed shrimp production reached a staggering 5.17 million tons in 2022, valued at USD 46.9 billion, with the global shrimp market projected to experience exponential growth in the coming years, fueled by escalating demand from the food industry [1,4]. However, notwithstanding their paramount importance, shrimp farming confronts formidable challenges primarily stemming from disease outbreaks, environmental fluctuations, and due to improper utilization of chemicals [7,8]. Shrimp aquaculture also experiences economic losses of over USD 3 billion every year due to diseases, and their spread is enhanced by the international trade of cultured shrimp, posing significant threats to the long-term sustainability of the industry [9,10]. The initial efficacy of antibiotics in mitigating shrimp infections has been compromised by the emergence and proliferation of antibiotic-resistant bacteria, and the dissemination of antibiotic-resistant genes from shrimp farms to the surrounding environment has led to potential risks to the well-being of humans [11,12]. Consequently, various environmentally conscious approaches have been investigated as substitutes for conventional disease management techniques in shrimp aquaculture. There has been growing interest in the utilization of phytochemicals, bacteriophages, immunization protocols (vaccinations), feed additives (such as probiotics and synbiotics), and cutting-edge genetic methodologies to control shrimp diseases [13,14]. An emerging field of scientific research called “epigenetics” has also offered great promise for preventing and managing infectious diseases in shrimp aquaculture [15,16].

Epigenetics focuses on the study of heritable changes in gene expression and gene expression patterns without altering the underlying DNA sequence. Epigenetic modulations encompass chemical alterations that occur in both DNA and proteins, ultimately modulating the accessibility of DNA to the transcription machinery [16,17]. Notably, processes associated with DNA methylation and histone modifications can directly orchestrate these epigenetic changes; additionally, non-coding RNA molecules have demonstrated the ability to influence gene expression patterns through post-transcriptional mechanisms and induce epigenetic changes [18,19]. Recently, there has been a growing interest in utilizing “targeted epigenetic modulations (purposefully changing the epigenome to obtain specific outcomes)” to enhance the biological capabilities of aquatic animals, including fish and shellfish [15,20]. The strategic deployment of such interventions holds tremendous promise in potentiating immune gene expression, fortifying resistance against diseases, and modulating the composition of the gut microbiome, thereby ultimately enhancing biological functions in shrimps [21,22]. Moreover, given the evidence on epigenetic inheritability among marine animals, the proper application of this approach can have a tremendous impact on the future of the shrimp farming industry [15,19,22]. Furthermore, ongoing advancements in epigenetic research, if coupled with technological innovations in the field, present exciting opportunities to develop targeted and tailored interventions for preventing infectious diseases in shrimp aquaculture.

Given this background, the primary objective of this comprehensive review is to elucidate the prospective application of epigenetics in the context of shrimp aquaculture for effective management and prevention of infectious diseases. It further underscores the importance of utilizing epigenetic regulatory mechanisms to ensure the long-term viability of the shrimp farming sector as a sustainable alternative to conventional disease control methodologies. Additionally, this review discusses the recent applications of epigenetic research, thereby presenting promising opportunities to develop targeted and precision-based interventions aimed at preventing infectious diseases in relation to shrimp aquaculture. The elucidation of such scientific developments bears profound implications for the sustainable growth of the aquaculture industry and global food security.

## 2. Current Status of Global Production and Challenges in Shrimp Aquaculture

Since the establishment of commercial shrimp farms in 1970, shrimp aquaculture production has experienced consistent growth with accelerated expansion. Its popularity and rapid momentum surged after 1990, and in 2007, shrimp aquaculture production surpassed fisheries production for the first time, accounting for a ratio of 50.2% to 49.8% [4]. Until 2008, farmed shrimp production stood at approximately 3.7 million tons, representing over 50% of the global shrimp market (Figure 1). In 2018, global farmed shrimp production reached 4 million tons, indicating a 3–5% increase from the previous year [23]. The production of shrimp has witnessed a remarkable surge, from less than 0.6 million tons in 1980 to over 5 million tons in 2022, signifying nearly a ten-fold upsurge [1,4,10]. Projections for global farmed shrimp production indicate a rise to 7.28 million tons by 2025, underscoring the pivotal role of shrimp aquaculture in meeting global food demand and the substantial growth of the industry throughout the years [24]. Currently, the majority of shrimp production is concentrated in East and Southeast Asia and Latin America, while a substantial portion of consumption occurs in developed markets such as the United States of America, the European Union, and Japan [25]. Presently, the leading global shrimp producers include China, Thailand, Vietnam, Indonesia, India, Mexico, Malaysia, Brazil, and the Philippines, underscoring the predominant concentration of the industry in the Asian region, accounting for 80% of global production [26]. Asia’s dominance in shrimp aquaculture is attributed to its favorable climate, extensive knowledge of shrimp farming, and to its well-established infrastructure. The region’s affordability of labor, market demand, and government support further contribute to its leading position in global shrimp production [27]. Among the various shrimp species cultivated, the dominant farmed species comprises white-leg shrimp (*Litopenaeus vannamei*) and the giant tiger shrimp (*Penaeus monodon*), which collectively capture over 90% of the market share (Figure 2). However, *L. vannamei* has emerged as the preferred choice for cultivation over *P. monodon* in numerous regions due to factors such as the availability of special pathogen-free broodstock, lower protein requirements, production costs, tolerance to diverse water parameters, distinctive flavor profile, and nutritional value. Additionally, *L. vannamei* demonstrates adaptability to a broad range of salinity levels and temperatures, enabling convenient cultivation practices in inland waters [28,29,30].

Throughout the history of shrimp aquaculture history, the industry has grappled with global pandemics and diseases caused by pathogens, but it has demonstrated adaptability and resilience in coexisting with these challenges while expanding production [26,31]. Even in the present day, pathogens remain the most devastating threat to the shrimp aquaculture industry, resulting in significant economic losses and causing drawbacks to the development of the industry [32,33]. The unrestricted international trade of shrimp and related materials, including broodstock, has facilitated the global expansion of shrimp diseases [34]. Among different pathogens, viruses emerge as the most destructive threat to farmed shrimp [35,36]. A multitude of viral organisms capable of causing diseases have been identified, with white spot syndrome virus (WSSV) standing out as the most pervasive pathogen afflicting farmed shrimp on a global scale. The disease, known as white spot disease (WSD), was first reported in cultured kuruma shrimp in China and Taiwan in 1991/1992, rapidly spreading to other shrimp-farming countries in Asia and reaching Latin America by 1995 [37,38]. The cumulative economic losses due to WSD over the past two decades are estimated to be around USD 15 billion globally, with WSSV causing approximately two-thirds of the annual economic loss, equivalent to approximately USD 1 billion, and a 15% reduction in global shrimp production [39]. Among other viral diseases, the recently identified infectious myonecrosis disease (IMND) caused by infectious myonecrosis virus (IMNV) in *L. vannamei* in Brazil in 2002 has rapidly spread to other shrimp farming regions through the importation of broodstock materials, and current experimental studies have shown that all other commercial shrimp species, including *P. monodon* and *P. stylirostris*, are susceptible to IMNV infection [40]. In Brazil alone, IMNV has resulted in financial losses of approximately USD 440 million between 2002 and 2005, and by the end of 2011, estimated losses in Brazil and Indonesia exceeded USD 1 billion [41]. Furthermore, there is evidence of co-infection of IMNV with WSSV among *L. vannamei* shrimp populations, which makes the situation even worse [42]. 

**Figure 1 genes-14-01682-f001:**
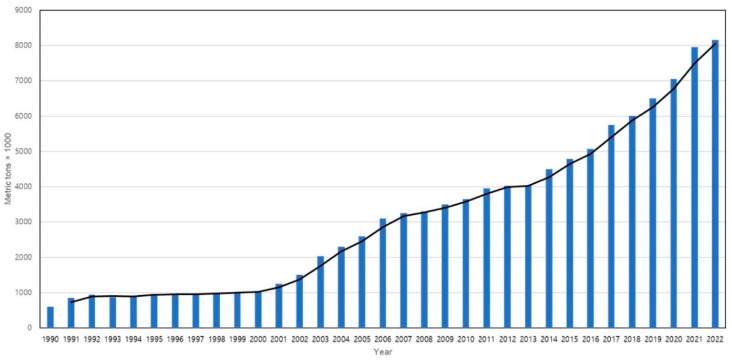
Global shrimp production (in metric tons) trend during 1990–2022 [43].

**Figure 2 genes-14-01682-f002:**
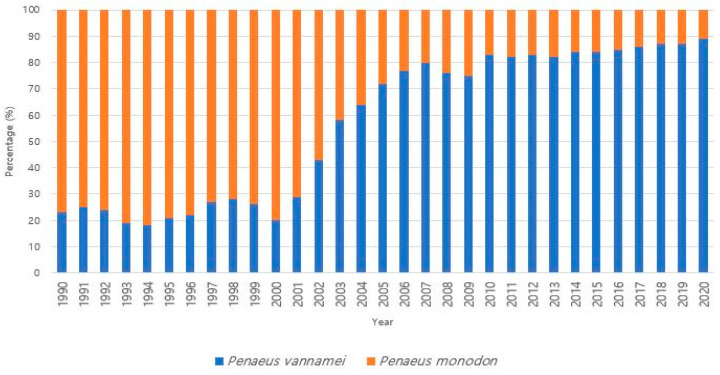
Global shift in production (in percentage) from black tiger shrimp to white-leg shrimp [43].

Apart from viral infections, a bacterial disease called acute hepatopancreatic necrosis (AHPN), mediated by *Vibrio* spp., has significantly impacted the global shrimp industry in recent years. AHPN, previously known as early mortality syndrome (EMS), was first reported in China in 2009 and subsequently spread to Southeast Asia (Vietnam, Malaysia, Thailand, Philippines, and Bangladesh) and South America (Mexico and the USA) [31,44,45,46]. Since the 2010s, AHPN has caused production declines in major shrimp-producing countries in Asia, resulting in substantial income losses estimated at over USD 10 billion, with annual losses reaching approximately USD 1 billion [10,47]. The disease’s ability to cause mass mortalities in shrimp populations has led to disruptions in the global supply for consumption, impacting the global seafood market and potentially leading to higher prices for consumers [48]. 

Shrimps also encounter multifaceted challenges to maintaining a balanced gut microbiota, which plays a pivotal role in modulating immune and cellular functions [49]. Several factors, encompassing water chemistry, feed composition, and antimicrobial usage, can disrupt the delicate equilibrium of the gut microbiome, resulting in recurring encounters with pathogenic invasions [50]. Additionally, fluctuations in environmental conditions, such as temperature and salinity, exert significant impacts on shrimp’s immune and physiological functions [51,52]. Moreover, the presence of immunosuppressive agents, including pollutants, within their habitats can impair the biological functions of these animals [53]. Also, the limited genetic diversity, the absence of advanced breeding techniques, and the challenges posed by genetic constraints, such as inbreeding depression in commercial shrimp populations, have hindered the development of superior shrimp strains with desirable traits [54,55]. 

With the intensification of shrimp aquaculture, the risk of rapid disease dissemination is amplified, impeding the implementation of effective control measures. As a countermeasure, recent scientific advancements have found ways to enhance the biological capabilities of shrimps. In recent times, the utilization of bacteriophages, vaccines, immune stimulants, and feed additives have emerged as viable alternatives to combat infectious diseases in the shrimp farming industry. 

## 3. Overview of Current Shrimp Disease Management Strategies

Throughout its historical trajectory, the shrimp aquaculture industry has encountered recurrent global pandemics, with diseases exerting a pivotal influence on its evolution [37,56,57]. While diverse pathogenic ailments persistently present challenges, the industry has demonstrated remarkable resilience in managing and adapting to disease outbreaks during the preceding two decades, coinciding with a substantial amplification in production. This resolute response portends that the industry will persistently acquire knowledge and devise effective strategies to effectively confront and coexist with diseases as its global expansion persists [56]. Consequently, numerous approaches have been embraced to surmount the obstacles imposed by pathogens and are presently under inspection. These methodologies encompass the utilization of antibiotics, bacteriophages, vaccines, immune stimulants (such as algal extracts), and feed additives (including probiotics, prebiotics, and synbiotics) to combat infectious diseases in shrimp. Recent empirical findings substantiate the efficacy and safety of these modalities as viable solutions for disease control in the shrimp aquaculture industry.

The foremost approach employed for shrimp disease prevention is the utilization of antimicrobials and antibiotics. Nevertheless, toward the conclusion of the 20th century, practical impediments emerged in the application of antibiotics due to the acquisition of tolerance by microorganisms to these chemical agents [11,56]. Consequently, the adoption of antibiotics and chemical agents to combat bacterial infections in shrimps has facilitated the dissemination of antibiotic-resistant bacteria, thereby engendering environmental hazards and the deposition of drug residues in their tissues [58,59,60]. Recent investigations also showcased a higher prevalence of antibiotic residues in farmed shrimp as compared to their wild-caught counterparts, highlighting the pervasive and unregulated usage of antibiotics in shrimp farming to augment production [60,61,62,63]. Furthermore, the genetic elements conferring drug resistance in numerous antibiotic-resistant bacteria can be transmitted through horizontal gene transfer (HGT) to other pathogenic microorganisms via contaminated aquaculture feed, thus posing a considerable health concern for human populations [58,64]. Shrimp pathogens, particularly those belonging to the *Vibrio* genus, have demonstrated such resistance to antibiotics commonly employed in aquaculture settings. The ingress of antibiotic-resistant strains into shrimp farms can transpire via seawater, as effluents laden with antibiotics and chemicals are discharged into marine environments. On the other hand, excessive employment of antibiotics and chemicals in shrimp aquaculture ponds contributes to the degradation of water quality, culminating in immune suppression induced by stress events [61]. Considering the mounting peril of climate change, the prevalence of antibiotic resistance in pathogens is anticipated to escalate, imposing repercussions on both aquatic organisms and human well-being.

Vaccination serves as a vital prophylactic measure in aquaculture, particularly in marine vertebrates (e.g., fish), to augment the host’s immunological defenses against infectious agents. Given the existence of an adaptive immune system and the potential for immune memory, vaccines exert markedly greater effectiveness in these organisms [65,66,67]. Nonetheless, invertebrates such as shrimp lack an adaptive immune system, thereby raising the pertinent query: “Can vaccination be efficacious in shrimp?”. However, diligent research endeavors have been directed toward the development of DNA vaccines for shrimp employing biodegradable carriers such as chitosan, and encouraging outcomes from these investigations demonstrate that oral administration of chitosan in *P. monodon* confers immune enhancement and safeguards against the WSSV [68,69]. Liposome-based vaccines, structurally resembling cellular membranes, have been explored in shrimp aquaculture, and notably, the administration of a liposome-based recombinant VP28 vaccine has yielded impressive survival rates against WSSV in *Marsupenaeus japonicus* [70]. Additionally, the exploitable potential of formalin-inactivated *Vibrio*, administered orally to *Fenneropenaeus merguensis* postlarvae, has elicited noteworthy protection against subsequent *Vibrio harveyi* infections [71]. However, the development of a commercial shrimp vaccine represents a multifaceted process that necessitates the requirement of intricate comprehension of the presence of shrimp’s unique “adaptive-like immunity.” Also, developing vaccines for different pathogens can be complex and time-consuming, and there is a risk of reducing vaccine efficacy over time [72]. Moreover, production costs, logistical challenges, and potential environmental impacts can limit their effectiveness. 

Bacteriophages (phages) have emerged as promising therapeutic agents in the field of aquaculture, owing to their remarkable efficacy and safety in combating pathogenic bacteria. These viral predators offer an environmentally friendly approach, capitalizing on their innate ability to selectively target and eliminate specific bacteria while sparing non-target microbiota due to their restricted host range. The deliberate administration of these phages (known as phage therapy) holds great potential for the management of bacterial diseases in aquaculture [73,74]. The delivery of phages can be achieved through various means, including injection, dietary administration, or their incorporation into rearing systems [75]. Among these, dietary administration is particularly favored as it facilitates the treatment of large populations and offers protection against intestinal infections. In recent times, noteworthy successes have been documented in the application of phage therapy for the treatment of AHPND caused by *Vibrio parahaemolyticus* in shrimp aquaculture [76]. Additionally, the utilization of phage cocktails, consisting of combinations of multiple phages, has demonstrated synergistic effects and superior efficacy compared to individual phages against *V. parahaemolyticus* and *V. harveyi* in artemia shrimp models [77]. Furthermore, phages have displayed promising anti-biofilm activities against shrimp pathogens, with specific phages disrupting bacterial protein machinery and thereby implying the existence of encoded antimicrobial compounds [78]. However, it is imperative to acknowledge the possibility of bacterial resistance to phages, arising from receptor loss on the cell membrane as well as the presence of restriction enzymes and CRISPR systems (clustered regularly interspaced short palindromic repeats, a natural defense system that allows bacteria to store and later use genetic information from viruses they have encountered) [79,80,81]. Also, horizontal gene transfer has been observed in some lytic phages, enabling the dissemination of antibiotic-resistant genes, and the occurrence of virulent mutants among temperate phages may complicate treatment regimens as they possess the capacity to revert to lysogeny [81]. Thus, it is essential to carefully consider the development of phages or tailored phage cocktails that are capable of overcoming the above obstacles to ensure both efficacy and safety.

In recent years, there has been a growing interest in the utilization of immunostimulants as a viable alternative to antibiotics for boosting the immune system of animals. Immunostimulants encompass naturally occurring and/or synthetic substances derived from diverse sources, including bacteria, algae, animals, and plants [82]. Within the shrimp aquaculture industry, the efficacy and multifunctionality of immunostimulants, such as polysaccharides derived from seaweeds, have been extensively investigated. Notably, brown seaweeds have been identified as a valuable source of bioactive compounds, including fucoidan and alginate, which possess immunostimulant properties and confer resistance against shrimp pathogens [83]. Moreover, plant extracts from various herbs (such as *Eclipta alba*, *Aegle marmelos*, and *Cyanodon Dactylon*) have demonstrated antiviral and immunostimulant activities against viral organisms like WSSV [84]. These scientific findings underscore the perspectives of immunostimulants derived from diverse sources as effective alternatives for disease management in the shrimp farming industry. Furthermore, the utilization of feed additives (including probiotics, prebiotics, and synbiotics) has caught the eye in recent times as a promising strategy for enhancing shrimp biological functions. Probiotics, which encompass bacteria-based products, have showcased abilities to promote health, prevent diseases, and control the proliferation of pathogenic organisms [85]. Notably, several bacterial species, such as Bacillus, *Vibrio*, and lactic acid bacteria, have been introduced as probiotics in shrimp farming, demonstrating their potential to enhance the overall health of shrimp populations [86,87]. On the other hand, prebiotics are indigestible fibers capable of fostering the growth and activity of beneficial bacteria within the gut, thereby promoting shrimp health. Several prebiotics have been introduced to shrimp farming, including inulin, fructooligosaccharides (FOS), and mannan oligosaccharides (MOS), which are derived from plant sources and fibers [88,89]. Synbiotics (representing a combination of prebiotics and probiotics) exhibit a synergistic relationship by facilitating the gut microbiome growth while enhancing the immune resistance in shrimps against pathogens, as demonstrated by the supplementation with *Lactiplantibacillus plantarum* and galactooligosaccharide [90]. The selection of specific prebiotics and probiotics in synbiotic formulations is based on their ability to enhance the proliferation of beneficial bacteria and confer overall benefits to the host. However, overall, it is important to acknowledge that the efficacy of these control methods may exhibit their positives, negatives, variabilities, and inconsistencies. Also, the possibility of these archived traits may pose challenges in terms of their stability and heritability over multiple generations, thereby raising significant concerns regarding the economic sustainability of the shrimp aquaculture industry.

## 4. The Concept of Epigenetics, Epigenetic Inheritance, and Its Potential Effect on Disease Management in Shrimp

Shrimp aquaculture confronts various challenges, particularly the rise in diseases that exert a significant impact on both economic and production aspects. Conventional disease management strategies, such as the implementation of antibiotics and vaccines, possess inherent limitations and potential drawbacks. Consequently, researchers and industry professionals have redirected their focus toward alternative approaches, notably the potential of utilizing epigenetic modulation to enhance disease resistance and bolster aquatic animal health [16,91,92]. Epigenetic modulation, a noteworthy method, encompasses the modification of gene expression patterns devoid of alterations to the underlying DNA sequence [93]. It is important to acknowledge that epigenetic regulatory mechanisms are part of a complex interplay that works in conjugation with other genetic, environmental, and physiological factors that collectively influence an organism’s biological functions (such as development, adaptation, and response to environmental stimuli) [16,94,95]. Epigenetic modulations are governed by three fundamental mechanisms, namely DNA methylation, histone modifications, and non-coding RNA regulation [96,97]. In the context of shrimp disease management, epigenetic modulation offers numerous advantages. 

Firstly, it furnishes a mechanism for swift and reversible adaptation to dynamic environmental conditions and pathogenic challenges. It has been demonstrated that shrimp, like other animals, possess the capacity to potentially modify their epigenetic marks, thereby activating immune-related genes that have the potential to augment disease resistance, which enables shrimp to mount efficacious defenses against pathogens [15,22,98]. Secondly, epigenetic modulations can influence the expression density of genes implicated in immune response pathways, thereby enabling shrimp to fine-tune their immune response and counteract diseases [98]. Hence, it may serve as a potential avenue for enhancing the shrimp’s immune system to obtain desirable outcomes without reliance on external agents, thus mitigating the risks associated with side effects and environmental contamination that are frequently linked with conventional disease management strategies.

An enthralling facet of epigenetic modulation lies in the prospect of “transgenerational epigenetic inheritance”, whereby these epigenetic changes are transmitted from one generation to the next. Transgenerational epigenetic inheritance takes place when the resetting of germline epigenetic marks is disrupted or circumvented, allowing the persistent transfer of these acquired epigenetic modifications [99]. This phenomenon has garnered extensive attention in investigations of other organisms, inciting interest in its applicability to shrimp disease management. Epigenetic inheritance furnishes the potential for “priming” future generations of shrimp, rendering them more resistant to diseases [98,100]. This facet of epigenetic modulation aligns harmoniously with the principles of sustainable aquaculture and also mitigates reliance on external interventions. Through exposure to specific environmental cues or epigenetic modifiers, it may be plausible to induce heritable epigenetic changes that enhance the biological functions of these animals. However, the mechanisms and extent of epigenetic inheritance in shrimp remain incompletely understood and necessitate further exploration. Studies aimed at identifying specific epigenetic marks associated with disease resistance and assessing their stability across generations will yield valuable insights into their applicability. 

## 5. Types of Epigenetic Modifications

### 5.1. DNA Methylation

In the field of epigenetics, DNA methylation emerges as the most extensively investigated phenomenon in the current research landscape, given its ability to serve as a stable and heritable epigenetic mark that can influence gene expression patterns [101]. DNA methylation encompasses the addition of a methyl group to nucleotides within a DNA strand, facilitated by DNA methyltransferase (DNMTs) enzymes. These DNMTs associated with DNA methylation are divided into maintenance DNMTs (*DNMT1* and *DNMT2*) and de novo DNMTs (*DNMT3*), where *DNMT1* is the most abundant and responsible for the management of existing DNA methylation patterns (they are capable of recognizing hemimethylated DNA and the addition of methyl groups to the newly synthesized DNA strand to preserve the methylation pattern) while *DNMT2* is associated with the methylation of RNA [101,102,103]. De novo DNMTs (including *DNMT3A* or *DNMT3B*) are involved in the establishment of new DNA methylation patterns and are capable of methylating previously unmethylated DNA regions, hence showcasing pivotal roles in biological processes such as gametogenesis [104]. In shrimp, the presence of all three DNMTs (*DNMT1*, *DNMT2*, and *DNMT3*) has been identified, suggesting the possible presence of DNA methylation in this organism [105]. 

Based on previous scientific research, it has been proven that DNA methylation occurs across diverse genomic regions, including promoters and enhancers regions, ultimately influencing various gene expression outcomes dependent on the level of methylation [106,107]. Notably, methylation of promoter regions is often associated with gene silencing, leading to the inhibition of further gene expression where the impact of DNA methylation on enhancer regions is context-dependent, resulting in either gene expression inhibition or enhancement [108,109]. There is a growing body of evidence highlighting significant differences in DNA methylation patterns between invertebrates (such as shrimps) and vertebrates. Also, DNA methylation among different species might be evolution-dependent, showcased by the high variability of methylation dynamics between vertebrate taxonomic groups such as mammals and reptiles [110]. However, in comparison to vertebrate animals, invertebrates have lower levels of DNA methylation in their genomes. Though a significant proportion of invertebrates exhibit DNA methylation levels ranging from 20% to 40%, shrimps, in contrast, display a comparatively low DNA methylation level of approximately 2% relative to their genome size [110]. On the other hand, vertebrates predominantly exhibit prominent CpG site methylation (characterized by regions containing a cystine nucleotide followed by a guanine nucleotide) across their genomes. However, the CpG islands that contain a high frequency of CpG dinucleotides (which has shown to coincide with gene promoter regions) are typically unmethylated and correlate with active gene expression, allowing the transcription machinery to access the DNA and initiate gene transcription, and this process has been shown to play a critical role in the development, cellular differentiation, and maintenance of cellular integrity in vertebrates demonstrated by previous studies in humans and mice [111,112]. Conversely, invertebrates (including insects, mollusks, and crustaceans) exhibit variable frequencies of CpG sites or lack CpG islands altogether, showcasing that CpG methylation is not the only factor explaining DNA methylation across these species, suggesting the utilization of alternative forms of methylation (named as non-CpG methylation) such as CpA or CpT sites [110,113]. Furthermore, DNA methylation in vertebrates, such as fish, birds, and mammals, has shown more pronounced tissue and cell type specificity and inter-individual differences, sculpting gene expression patterns and ensuring proper cellular functions [110]. Vertebrates finely regulate gene activities by selectively methylating and/or demethylating specific genome regions, contributing to this tissue specialization or cell differentiation, and in invertebrates, this has not been the case even though they have versatile ranges of potential methylation regions and has not shown any significant tissue specificity or inter-individual differences in DNA methylation patterns, where they are often correlated to the developmental stages and/or environmental stimuli, signifying a dynamic and adaptive nature that allows them covey rapid adjustments in respective gene expression patterns [110]. Given this degree of diversity, it has been suggested that invertebrates may be attributable to the utilization of diverse mechanisms beyond DNA methylation for gene expression and cellular processes.

In addition to variations in methylation types, noteworthy distinctions exist in the genome-wide distribution of DNA methylation between vertebrates and invertebrates. Vertebrates typically exhibit a global pattern of DNA methylation throughout the genome, while invertebrates display a mosaic pattern characterized by regions of heavy methylation interspersed with nonmethylated regions, which is more variable and species-specific [110,114]. These mosaic methylation patterns likely play a role in the target-specific regulation of genomic regions or functions, suggesting potential divergence in the regulatory mechanisms associated with DNA methylation in these organisms. Another notable difference between invertebrates and vertebrates lies in gene body methylation, which is prevalent in vertebrates and plays a significant role in alternative splicing; conversely, the majority of invertebrates exhibit lower levels of gene body methylation (Figure 3) [110,115,116]. The absence or reduced presence of gene body methylation in invertebrates suggests the employment of alternative molecular strategies, such as histone modifications, chromatin remodeling, or other epigenetic mechanisms, to achieve similar regulatory outcomes as gene body methylation in vertebrates. Additionally, DNA methylation in vertebrates serves a crucial role in controlling the activities of transposable elements (TEs) within the genome. By adding a methyl group to the DNA sequence of TEs, DNA methylation can manipulate defense mechanisms to reduce their mobility and transcription, thereby preserving genomic integrity [117]. In contrast, invertebrates like shrimp typically rely on mechanisms such as small RNA-based silencing pathways involving small interfering RNAs (siRNAs) and piwi-interacting RNAs (piRNAs) to control post-transcriptional or transcriptional gene silencing highlighting the diverse strategies employed by these organisms [118,119].

### 5.2. Histone Modifications

Histone modifications are fundamental epigenetic mechanisms that intricately regulate gene expression patterns. Histones, crucial protein molecules involved in DNA packaging within chromatin, undergo chemical alterations that dynamically modulate DNA accessibility to regulatory proteins. Among animals, a diverse repertoire of histone modification programming has been unraveled, encompassing methylation, acetylation, ubiquitination, phosphorylation, sumoylation, and ADP-ribosylation [120]. While the role of histone modifications in vertebrates (including mammals, birds, reptiles, amphibians, and fish) has been extensively studied, providing a comprehensive understanding of their impact on gene regulation, and in invertebrates, such as crustaceans, this epigenetic mechanism has garnered less attention, consequently leaving its significance unexplored.

Compared to invertebrates, vertebrates generally exhibit a more complex chromatin structure where they are organized into distinct domains, including euchromatin and heterochromatin, as well as higher-order structures like nucleosomes and chromatin loops. In contrast, invertebrate chromatin possesses a relatively simpler structure with fewer defined domains and some higher-order structures [121,122,123]. It is thought that these structural disparities between vertebrates and invertebrates might be attributed to the differences in evolutionary history, genetic mechanisms, and the specific necessity of the regulatory mechanisms associated with histone modifications [124]. Among different mechanisms used, histone acetylation is the most extensively studied histone modification that relies on the enzymatic actions of histone acetyltransferases (HATs) and histone deacetylases (HDACs), which exert pivotal roles in regulating a wide range of physiological phenomena including cell differentiation, proliferation, metabolism, senescence, immune response, and programmed cell death [125,126]. Recent research in vertebrates and invertebrates has shown that histone acetylation primarily occurs on four histone proteins (H2A, H2B, H3, and H4), with H3 and H4 exhibiting the highest levels of acetylation [127,128]. Notably, acetylation of histone H3 can modulate ATP-dependent chromatin remodeling, thereby contributing to the inflammatory response, while histone H4 acetylation influences chromatin transcription and replication [129,130,131]. Concerning shrimps, all four histone proteins capable of acetylation have been identified, revealing their involvement in possible regulatory mechanisms [132]. Moreover, a recent study, directed using the artemia model, indicated the presence of a significantly higher level of H3 acetylation in the treatment group (parents and their progeny cysts) compared to its control group following phloroglucinol treatment, where H4 acetylation levels were significantly lower in the treatment group compared to the respective controls [15].

Histone methylation and/or demethylation represent another fundamental process implicated in the modulation of epigenetic alterations. The methylation of histones is catalyzed by histone methyltransferases (HMTs), while demethylation is executed by histone demethylases (HDMs), and their collective effort, along with the recruitment of various cofactors and chromatin modifiers, governs the precise patterns of this process, thus contributing to the regulation of diverse cellular functions. Although histone methylation/demethylation is a conserved histone modification method found in both vertebrates and invertebrates, distinct variations and functional outcomes can be expected depending on the context of this process. In contrast, it has been proven that histone methylation in vertebrates (such as mammals) can lead to gene activation or repression, depending on the specific site and degree of methylation [133]. Notably, the methylation of histone H3 (at the H3K4 location) has demonstrated the exclusive presence of three types of methylation, namely H3K4me1, H3K4me2, and H3K4me3 in genes and promoter regions. The H3K4me3 and H3K4me2 methylations primarily occurred in promoters, particularly in the 5’ end of transcribed regions, while H3K4me1 is depleted in promoters but enriched within transcribed regions [134,135,136]. Concerning shrimps, recent scientific evidence on the artemia model has shown conserved histone H3K4 methylation patterns with increased levels of H3K4me3, H3K4me1, and H3K4me2, indicating enhanced trimethylation at H3K4 sites similar to that of vertebrates. However, the probability of inheriting the H3K4 methylation has not been constant, leading to a progressive decline in the respective succeeding generations. The same research study also demonstrated the methylation occurrence in K9 and K27 positions in the H3 protein (Figure 4) [15]. Another study on *Marsupenaeus japonicus* has shown the potential utility of histone H3 methylation in enhancing trained immunity, suggesting its involvement in possible epigenetic inheritance [137].

Phosphorylation, ubiquitination, and sumoylation are additional histone modifications that play important roles in gene expression regulation by modulating chromatin structure and recruiting regulatory proteins. These modifications have been extensively characterized in vertebrates, such as mouse, and invertebrates, such as common fruit fly (*Drosophila melanogaster)* and oysters (*Crassostrea virginica*), while our understanding of their significance in crustaceans remains limited [138,139,140,141]. In the model organism *D. melanogaster*, phosphorylation of histone H3 at specific serine residues (H3S28 and H3S10) is linked to both gene activation and repression, respectively, at enhancers and promoters [142]. Additional investigations into the role of the chromatin-remodeling factor SMARCA1 in *Drosophila* neural stem cell differentiation also revealed that the involvement of histone H3 phosphorylation (at H3S10) is linked to gene activation and repression [143]. A similar type of serine phosphorylation in the histone molecule H2A has been observed in oysters (*C. virginica*) following exposure to brevetoxin [140]. Moreover, histone ubiquitination in *Drosophila* has been implicated in transcriptional activation and repression. For example, histone H2B ubiquitination has been linked to the transcriptional activation of certain genes (such as heat shock proteins), while ubiquitination of histone H2A is associated with gene silencing [144,145,146]. Among the crustaceans, unique H2A H4 phosphorylations have been detected in the mitten crab (*Eriochier sinensis*), and in mantis shrimps (*Oratosquilla oratoria*), H3 and H2A have been detected in the nuclei of spermatozoa [147,148]. Furthermore, a recent study on *L. vannamei* infected with AHPND-causing pirAB toxins revealed the occurrence of changes in H3 phosphorylation (at H3S10), which led to hemocyte apoptosis events [149]. Additionally, in the Chinese white shrimp (*Fenneropenaeus chinensis*), increased levels of H3 phosphorylation have been highlighted following WSSV infections [150]. However, it is imperative to acknowledge that the exploration of histone modifications in crustaceans is a relatively new field of study, requiring additional scientific investigations to elucidate the intricate mechanisms that govern gene expression and cellular processes in these organisms.

**Figure 4 genes-14-01682-f004:**
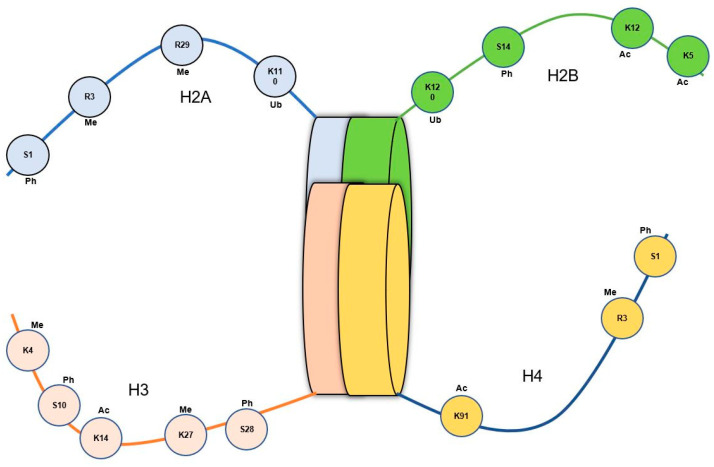
Schematic representation of post-translational events associated with H2A, H2B, H3, and H4 proteins, highlighting their modification types in epigenetic regulation in animals. Notably, histone methylation (Me), histone phosphorylation (Ph), histone acetylation (Ac), and histone ubiquitination (Ub) are the common mechanisms involved in histone modification. In the context of shrimps, previous investigations have identified specific histone modification patterns, including H3K4, H3K9, and H3K27 methylation; H3K14 and H4 acetylation; and H3S10, H3S28, H2A, and H4 phosphorylation [15,148,149,150]. These modifications play critical roles in regulating gene expression and contribute to various biological processes in shrimps.

### 5.3. Non-Coding RNA

Non-coding RNAs (ncRNAs) have emerged as pivotal players in the realm of epigenetic modifications, encompassing both vertebrates and invertebrates. These ncRNAs, which do not code for proteins, are classified into two categories, housekeeping and regulatory ncRNAs, where, based on their size, regulatory ncRNAs are further categorized as short-chain ncRNAs (including siRNAs and miRNAs) and long non-coding RNAs (lncRNAs) (Figure 5) [96,151,152]. These ncRNAs have demonstrated crucial roles in various processes (including stress responses and immune functions), particularly involving the mechanism called “RNAi (RNA interference)”, which is a biological process that involves the silencing of gene expression by targeting specific mRNA molecules for degradation and/or translational repression across various organisms [153,154,155]. While there are shared regulatory principles of ncRNAs in vertebrates and invertebrates, the evolutionary and functional divergence among these groups has given rise to some peculiar differences [156,157]. Furthermore, the repertoire and functional significance of these ncRNAs can vary even across the invertebrate taxa, reflecting their adaptations to a wide range of lifestyles.

Among short-chain ncRNAs, the endogenous small RNA molecules that are transcribed from specific genomic regions, commonly known as microRNAs (miRNAs), have garnered significant attention in animals due to their ability to post-translational regulation genes through partial binding to the 3′ untranslated region (UTR) of messenger RNA (mRNA) [158]. This binding can either promote, inhibit, or change the translation, leading to changes in various biological processes such as embryonic development, immune responses, neuronal development, and the pathogenesis of diseases (including cancer and neurodegenerative disorders), as demonstrated by various studies from vertebrates and invertebrates [159,160,161,162]. In the context of shrimps, the first set of miRNAs (35 miRNAs) was identified from *M. japonicus* hemocytes in 2011, 11 of which showed homology to miRNAs found in other arthropods [163]. Subsequent research has focused on miRNAs associated with defense response regulation in *P. monodon* against WSSV pathogen, revealing specific miRNAs (including miR-7, miR-965, and miR-12) that play crucial roles in virus–host interactions, affecting viral gene expression, replication, immune-related gene targeting, apoptosis, and immune recognition [164]. Similarly, in the case of *V. alginolyticus* infection in *M. japonicus*, differentially expressed miRNAs (including miR-100, miR-275, and miR-279) have been linked to innate immune responses involving phenoloxidase enzyme activation, apoptosis, and phagocytosis. Notably, miR-100 has played a pivotal role in regulating shrimp hemocyte apoptosis, promoting the anti-*Vibrio* immune response by influencing factors such as the extent of hemocyte count [165]. Furthermore, differentially expressed miRNAs, such as miR-210, miR-10b, and miR-193, have been associated with immune-related processes in AHPND caused by *V. parahemolyticus*, offering potential insights into immune responses [166]. Additionally, environmental factors such as copper exposure and heat stress have also been instrumental in the identification of novel miRNAs in shrimp [167,168]. 

Small interfering RNAs (siRNAs) are another type of ncRNA that have been identified as key regulators of potent antiviral defense mechanisms in both vertebrates and invertebrates. The siRNAs are double-stranded RNA molecules that are typically generated from exogenous sources, such as viral genomes or transposable elements. They guide the RNA-induced silencing complex (RISC) to respective complementary target mRNAs, leading to their degradation and translation prevention [169,170]. This process, known as post-transcriptional gene silencing, serves as a defense mechanism against foreign genetic elements, and in shrimp, it has been shown to play a crucial role in the immune defense against viral infections by effectively degrading viral RNAs and restricting their replication. In shrimp, such as *M. japonicus*, the ability to generate an antiviral siRNA (vp28-siRNA) in response to WSSV infection has been detected, and its production and function have been shown to coincide with the availability of RISC [171]. Considering the association of miRNAs and siRNAs with epigenetic regulation, these newly discovered ncRNAs hold great promise for disease management strategies in shrimp aquaculture.

Piwi-interacting RNAs (piRNAs) are another distinct class of non-coding RNAs (ncRNAs) that play pivotal roles in gene regulation and genome defense mechanisms. These small RNA molecules interact with piwi proteins, which are a subgroup of the Argonaute protein family. In comparison to other classes of small RNAs, such as miRNAs and siRNAs, piRNAs are primarily expressed in the germline and are involved in maintaining genome integrity [172,173]. Moreover, in both vertebrates and invertebrates, piRNAs are capable of epigenetic modulations through the silencing of transposable elements (TEs), which are mobile genetic elements capable of disrupting genome stability. While interacting with piwi proteins, piRNAs guide the recognition and targeting of complementary sequences within TEs, resulting in their transcriptional and post-transcriptional repression, and this mechanism is vital for preventing the harmful effects of TEs, such as genomic instability and germ cell dysfunction [174,175]. In invertebrates, the roles of piRNAs have been extensively studied, as they are more abundant and highly diverse in these taxa than in vertebrates, and it is hypothesized that the expansion and diversification of transposons in invertebrate genomes have driven the evolution of robust piRNA pathways as a self-defense mechanism [176,177,178]. Regarding shrimps, three piwi gene clusters and piRNA-like molecules have been identified; however, their role in epigenetic modulation remains to be explored [179,180,181]. The long non-coding RNAs (lncRNAs) are also a regulatory type of ncRNA, with essential regulatory functions, particularly in gene expression regulation at the transcriptional and/or post-transcriptional levels through interactions with DNA, RNA, and proteins. In shrimps, a large number of lncRNAs have been identified, revealing species-specific and tissue-specific expression among *L. vannamei*, *P. monodon*, and *Macrobrachium rosenbergii* [182,183,184]. 

Overall, the diverse repertoire of ncRNAs showcases the intricate mechanisms underlying shrimp biology and the possible utilization of these mechanisms for future applications in epigenetic modulation. Furthermore, utilizing the potential of these ncRNAs for disease management and selective breeding programs in shrimp aquaculture holds great promise for the sustainable and efficient production of these economically important crustaceans.

**Figure 5 genes-14-01682-f005:**
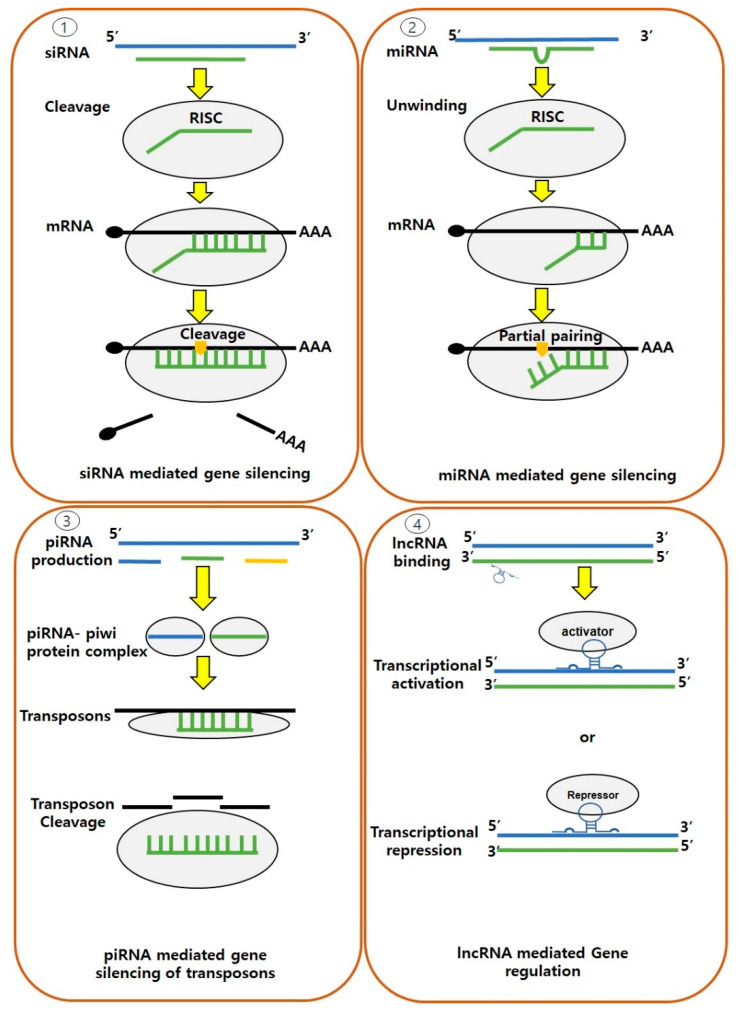
Schematic diagram depicting the elements of RNA implicated in epigenetic regulation across animal species. Notable examples encompass (**1**) small interfering RNA (siRNA), (**2**) micro RNA (miRNA), (**3**) piwi-interacting RNA (piRNA), and (**4**) long non-coding RNA (lncRNA). Each type of mechanism involves different types of regulatory molecules. In the context of shrimps, previous investigations have elucidated the existence of specific siRNA (such as vp28-siRNA) and miRNA (such as miR-100, miR-275, miR-7, miR-965) that combat viral and bacterial pathogens, including white spot syndrome virus (WSSV) and *V*. *alginolyticus* [164,165,171]. Overall, the participation of above non-coding RNAs (ncRNAs) in regulatory processes assumes a pivotal role in the modulation of gene expression and contributes to diverse biological processes in shrimps.

## 6. Applications of Epigenetics for Immunity Enhancement and Disease Management in Shrimp Aquaculture

Epigenetic modulation through targeted modification of epigenetic marks holds considerable promise for augmenting various molecular pathways in shrimps. Such modifications can be achieved through multifarious strategies, encompassing the administration of pharmacological agents and/or dietary supplements, environmental interventions, and gene-centric methodologies tailored to specifically target and modify epigenetic signatures. In the context of shrimps, recent investigations have demonstrated the possibility of epigenetic mechanism manipulation by utilizing the above-mentioned key strategies. For example, the efficacy of employing specific methyl donors with DNA methylation capacity, such as S-adenosylmethionine (SAM) and L-methionine or methionine hydroxy analogs, as dietary supplements have shown to modulate antioxidant capacity, cell proliferation ability, and protein synthesis, alternatively leading to better growth and defense performance in *L. vannamei* [185,186]. Conversely, dietary supplements enriched with folic acid, an essential nutrient for DNA synthesis, have exhibited ameliorative effects on the immunity-associated functions of *M. rosenbergii* [187]. Furthermore, in the artemia model, the application of plant-based phenolic compounds as a biological control treatment has evinced transgenerational effects, instigating an inherited enhancement in resistance against *Vibrio* that persists across subsequent generations, indicating a transgenerational response to the treatment. This heightened resistance has been postulated to be associated with augmented expression of innate immune genes, with DNA methylation serving as a regulatory factor governing the expression of these genes [15,98]. In addition, environmental interventions have been observed to yield promising outcomes in terms of inducing epigenetic modifications. Various factors, including temperature fluctuations, salinity, water quality, air availability, and feeding regimes, have been found to impact epigenetic changes and the functionality of immune-associated processes in animals, including shrimps. The extant evidence suggests that shrimps are capable of undergoing epigenetic changes and exhibiting transgenerational resilience in response to heat stress. For instance, the association between epigenetic mechanisms and temperature-induced tolerance to lethal heat shock and resistance against the pathogenic bacteria *V. campbellii* has been demonstrated in the artemia model. This study revealed altered levels of DNA methylation and increased acetylation of histones H3 and H4, accompanied by changes in heat shock protein 70 (HSP70) levels [22]. By building upon these intriguing findings, subsequent studies validated that exposure to other environmental stimuli, such as bioactive food components, could induce similar epigenomic modulation in artemia, resulting in the generation of robust phenotypes with resistance against *V. campbellii* over three successive generations [98]. Furthermore, alterations in water quality parameters, such as ammonia/nitrate levels, and exposure to environmental pollutants can instigate oxidative stress and potentially inflict damage to DNA in crustaceans, thereby potentially modulating epigenetic changes [188,189,190]. Additionally, air availability has been shown to influence DNA methylation patterns in kuruma shrimps, coinciding with changes in apoptosis and physiological responses [191]. Moreover, feeding regimes, entailing carefully devised plans encompassing diet composition, nutrient intake, and schedule, can exert significant impacts on the epigenetic changes associated with shrimp immunity and defense functions. 

In addition to the aforementioned approaches, gene-centric methodologies have recently emerged as effective means to elicit epigenetic alterations in animals. Among these methodologies, RNA interference (RNAi) stands out as the most commonly employed technique, entailing the attenuation or downregulation of specific target genes through the utilization of synthetic small-interfering RNA (siRNA) molecules [192]. Due to its versatility, RNAi has proven to be a valuable tool for inducing epigenetic modifications in shrimps. Notably, recent investigations have unveiled the potential of RNAi-mediated modifications in enhancing growth and development factors critical for mounting effective immune responses in shrimps. For instance, a study employed RNAi to silence the expression of m6A methyltransferase genes in *L. vannamei*, leading to diverse variations in DNA methylation patterns and subsequent alterations in molting-associated gene networks [193]. Moreover, applications of RNAi targeting immunity-related genes, such as the Kruppel-like factor 2 (KLF2) gene, are implicated in the immune responses of *L. vannamei* [194]. Also, the silencing of Toll and IMD genes has been shown to impact immune response pathways and influence transcription levels of antimicrobial peptides (AMPs) in the same species [195]. Similarly, RNAi-mediated knockdown of the “LGBP” gene in shrimps has induced expression changes in the prophenoloxidase (proPO) system and AMP production [196]. While these studies do not explicitly convey the occurrence of heritable epigenetic changes, they highlight the potential of RNAi as a mediator for delivering immune gene-associated alterations, thus holding significance for future investigations of epigenetic modulation to develop disease management strategies.

The revolutionary CRISPR/Cas9 gene editing approach has emerged as a powerful method for genome manipulation in animals [197]. While not directly considered an epigenetic modulation method, CRISPR/Cas9 can indirectly introduce epigenetic changes by altering the DNA configuration. The CRISPR/Cas9 process entails the precise targeting of specific genes and the introduction of mutations and/or insertions/deletions that can induce epigenetic changes. The majority of the previous reports on shrimps focused on the CRISPR/Cas9 systems related to growth and reproduction. In contrast, the role of CRISPR/Cas9 targeting genes involved in the regulation of molting in the oriental prawn (*Exopalaemon carinicauda*), resulting in mutations in the chitinase gene (named EcChl4) has led to alterations in chitin expression, which plays a pivotal role in the primary immune response of shrimp [198]. Conversely, CRISPR/Cas9 system-mediated mutation of the insulin-like peptide-encoding gene has impacted growth and development in various life stages, resulting in high mortality rates within the “gene knockout” populations in the same species [199]. Although the application of CRISPR/Cas9 technology is still in its nascent stages, these studies indicate the potential of employing it as a valuable tool for inducing subsequent epigenetic changes, particularly in the context of growth and development alterations in shrimps.

Withstanding these research findings, it is reasonable to posit that exploring the nature of epigenetics holds enormous potential for disease management in shrimp aquaculture and the augmentation of various facets of shrimp productivity. Manipulating certain epigenetic marks, such as DNA methylation, histone modifications, and non-coding RNAs, allows the alteration of gene expression patterns, thereby enhancing disease resistance and overall performance in shrimps. Furthermore, environmental factors, dietary interventions, and gene-centric methodologies can induce epigenetic changes that enhance shrimp health and optimize their physiological responses. Hence, unraveling the underlying mechanisms governing epigenetic regulation presents novel avenues for developing innovative strategies to revolutionize shrimp farming practices.

## 7. Challenges and Future Perspectives of Epigenetics in the Disease Management of Shrimp Aquaculture

The integration of epigenetic modulations in shrimp holds substantial promise, yet its practical implementation faces an array of intricate challenges and constraints. One primary hurdle lies in the demand for target-specific, finely tuned, and precise epigenetic interventions devoid of inadvertent off-target effects on other biological processes. This challenge is also influenced by our limited understanding of the shrimp epigenome. Given that the epigenetic landscape in shrimps is complex and is shaped by many different factors (such as environmental factors), deciphering the cross-talk among these mechanisms and their coordinated orchestration is essential for a comprehensive understanding of epigenetic regulation in shrimp [22]. Also, given the multi-genomic nature of shrimp genomes, implementing epigenetic modulations at a single unique site becomes intricate, as it may inadvertently influence neighboring regions, potentially yielding irreversible consequences for cellular functions [91,92,200]. Thus, the development of high-precision tools and methodologies capable of eliciting specific and localized epigenetic phenomena becomes paramount. Incorporating epigenetic modulations in shrimp aquaculture necessitates careful consideration of potential off-target effects, encompassing the emergence of drug resistance and perturbations in the gut microbiota composition, potentially instigating unforeseen consequences. 

The transgenerational inheritance of epigenetic changes can significantly impact shrimp populations. This phenomenon offers the potential to enhance shrimp health and disease resistance over multiple generations. However, prolonged or recurrent interventions might be required to sustain the desired epigenetic changes in animals, engendering logistical challenges regarding practicality and implementation feasibility as one-time interventions [15,201]. Moreover, the intricate interplay between shrimp physiology, metabolism, and gene expression patterns demands cautious evaluation, as epigenetic alterations could potentially perturb critical metabolic pathways essential for the animal’s health and well-being [92,191,202]. Hence, achieving comprehension of the complex interactions between epigenetic modifications and gene expression is essential, as it will enable the identification, validation, and development of targeted interventions (such as epigenetic targets associated with disease resistance) that optimize shrimp health and performance. In addition to scientific considerations, ethical obligations take center stage when conducting research involving live animals. Ensuring the welfare of the animals involved is paramount, and researchers must adhere strictly to ethical guidelines and regulations to minimize any harm or distress inflicted upon the animals during epigenetic studies. A responsible and ethical approach underpins the pursuit of epigenetic advancements in shrimp aquaculture, signifying a commitment to the well-being of the animal subjects involved.

Despite the complex challenges encountered in implementing epigenetic modulations in shrimp aquaculture, the immense potential benefits warrant substantial attention. The profound understanding and manipulation of epigenetic changes offer viable avenues to augment diverse biological functions, including immune regulation, growth, and reproduction [22,98,100,191]. The identification of epigenetic biomarkers offers a transformative approach to disease diagnosis in shrimp aquaculture. Targeted manipulation of these epigenetic marks can enhance the expression of pivotal genes in immune pathways, fortifying the shrimp’s capacity to combat infections, thus reducing the dependency on antibiotics and other chemical treatments. These markers, derived from specific epigenetic modifications, might provide early indications of disease susceptibility and resistance, allowing rapid detection and enabling timely interventions, thereby reducing economic losses. Further, incorporating epigenetic biomarkers into management strategies promotes shrimp health and overall aquaculture resilience, as it facilitates precise control of epigenetic changes to yield shrimp progenies with heightened disease resistance and enhanced immunity [15,22,97]. In this context, the application of techniques such as DNA demethylation and histone modification systems that allow precise manipulation of epigenetic marks is essential to enhance disease resistance and other desirable traits in shrimp populations. Also, comparative studies integrating transcriptomics and epigenetics in varied trait individuals are necessary; hence, they will aid in the identification of genes linked to desirable trait development in shrimps. 

The interaction between the environment and epigenetic processes opens new avenues for innovative approaches [203,204]. The concept of environmental management focuses on creating optimal conditions that trigger desired epigenetic responses. Unlike the traditional methods that often rely on external interventions, environmental management leverages natural processes within the shrimp’s physiological framework. Concurrently, the malleability of epigenetic changes in response to dietary components offers alternative prospects for nutritional programming in commercial shrimp aquaculture. Deliberate manipulation of the shrimp’s diet to target nutrients involved in epigenetic regulations presents an opportunity to induce favorable changes in gene expression patterns related to growth and reproduction. In the long term, this approach holds the potential to foster sustainable and cost-effective shrimp performance while mitigating the demand for high-cost feeding materials, thereby contributing to the development of an economically viable shrimp industry. The fisheries sector has already witnessed such advantages of epigenetics, exemplified by studies in rainbow trouts that demonstrated higher feed efficiencies and growth rates resulting from epigenetic interventions [205,206]. Furthermore, the integration of epigenetics into selective breeding programs holds immense promise for genetic enhancement in shrimp populations. Through comprehension of the epigenetic regulation of desirable traits, the incorporation of epigenetic information into breeding strategies accelerates genetic improvement in shrimp populations. This avenue culminates in the production of superior strains endowed with enhanced disease resistance, growth rates, and other economically crucial traits. The potential transfer of epigenetic marks observed in a mass population study of Pacific oysters (*Magallana gigas*) may also offer further insights into the interplay of epigenetics and genetics in enhancing desirable traits [207]. Finally, comprehensive consideration of the ecological implications of epigenetic modulations in shrimp aquaculture is imperative to identify sustainable and environmentally friendly approaches that promote shrimp health and minimize negative impacts on surrounding ecosystems. The integration of epigenetics in shrimp aquaculture necessitates an interdisciplinary approach, fostering synergistic collaboration among researchers, biotechnologists, and ecologists to usher in a new era of innovative, sustainable, and eco-conscious shrimp farming practices.

## 8. Conclusions 

In conclusion, the domain of epigenetics in shrimp aquaculture holds prodigious promise for disease management, optimized productivity, and the evolution of robust populations. The full realization of epigenetic modulation’s potential necessitates the surmounting of multifaceted challenges through ingenious research, precision-driven tools, comprehensive monitoring, and seamless integration of epigenetic insights into breeding programs and aquaculture methodologies. Recent advancements in tailored feed formulations, nutraceuticals, pharmaceutical agents, and state-of-the-art genome-centric technologies such as RNA interference (RNAi) and CRISPR-Cas9 have unveiled captivating avenues for augmenting shrimp capabilities via targeted epigenetic interventions. Nonetheless, the field of epigenetics remains in its infancy, unveiling a plethora of unanswered inquiries into the mechanistic intricacies that call for exploration in the foreseeable future. The profound exploration of epigenetic frontiers bears the potential to catalyze the establishment of an immensely efficient and resilient shrimp farming industry, engendering a symbiotic alliance between industry prosperity and ecological stewardship.

## Figures and Tables

**Figure 3 genes-14-01682-f003:**
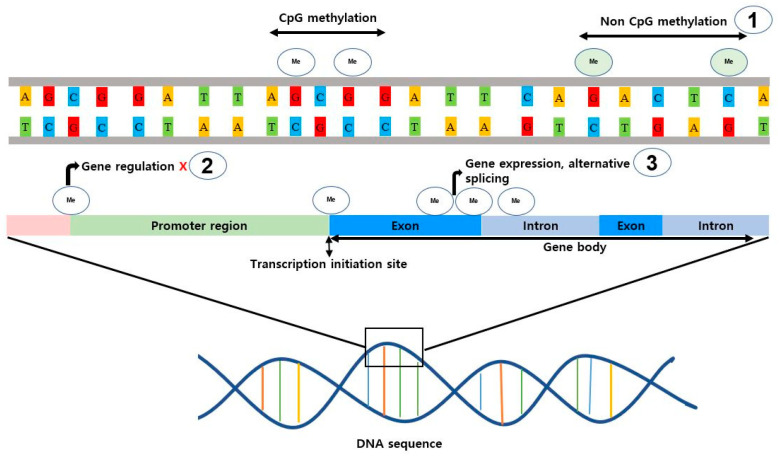
A comprehensive depiction of diverse categories and consequences of DNA methylation implicated in the epigenetic modulation of shrimps. DNA methylation encompasses two distinct regions: (1) CpG-rich domains, designated as CpG methylation, and regions lacking CpG enrichment, recognized as Non-CpG methylation. (2) Promoter region methylation exerts regulatory control over gene expression by inducing transcriptional inhibition. (3) Gene body methylation transpires within the intron-exon regions, contributing to alterations in gene expression levels and the emergence of “alternative splicing” phenomena.

## Data Availability

Not applicable.

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
