# Peer review of "Epigenetic Modulations for Prevention of Infectious Diseases in Shrimp Aquaculture"

_genes, 2023, doi:10.3390/genes14091682_

Round 1

Reviewer 1 Report

The present review paper entitled "Epigenetic Modulations for Prevention of Infectious Diseases in Shrimp Aquaculture" presents a compelling topic that addresses the potential application of epigenetic approaches in enhancing disease prevention strategies for shrimp aquaculture. This is a very interesting approach to explain the use of epigenetic mechanisms to bolster the shrimp's innate immune responses and resistance against pathogens as an intriguing concept with significant implications for the aquaculture industry. However, the present form of the manuscript needs minor revision. Below mentioned comments might help the authors to modify their manuscript.

1. Introduction:-

Line 33;- Instead of the data from 2020, authors should mention latest data from State of World Fisheries and Aquaculture (SOFIA) about the aquaculture production.

Line 47: Global shrimp production has reached 9.4 million tonnes by 2022 and farmed shrimp is 55% of it. Please mention latest data on the shrimp production as mentioned by FAO.

Line 70: Generally non-coding RNA molecules do not directly orchestrate epigenetic changes. Non-coding RNAs, such as microRNAs and long non-coding RNAs, influence gene expression through post-transcriptional mechanisms. Please correct accordingly.

Line 72: The term ‘targeted epigenetic modulation’ is not well understood from this statement. What is ‘burgeoning interest’?

2. Current Status of Global Production and Challenges in Shrimp Aquaculture:-

Provide latet data on global shrimp production trend.

Line 115: Please mention the reason why the mentioned nations become leading producer of shrimp. What are their strategies in production as common and resources?

Line 119: Prawn and shrimp are two different creatures. Please check whether to write tiger prawn or tiger shrimp.

Line 125: Though L. vannamei tolerate wide range of salinity, do they grow well in different salinity?

Line 172: It would be interesting to mention any case study with citation related to, how unrestricted international trade of shrimp and related materials, including broodstock, has facilitated the global expansion of shrimp diseases.

3. Overview of Current Shrimp Disease Management Strategies:-

Line 229: ‘Throughout its historical trajectory, the shrimp aquaculture industry has encountered recurrent 229 global pandemics, with diseases exerting a pivotal influence on its evolution’. Write a few examples and consequences of those pandemics on shrimp industry in a few lines with citation.

Line 251: How do genetic elements that confer drug resistance spread among antibiotic-resistant bacteria in aquaculture feed?

Line 262-268: Vaccination in aquaculture may not always be the most effective prophylactic measure, especially in certain contexts or for specific species. While it can be beneficial in augmenting the host's immunological defenses against infectious agents, there are some potential drawbacks and limitations to consider. It will be nice to include an explanation of the drawbacks of vaccinations in aquaculture in short.

Line 279-391: Many of the terminologies in this paragraph seems like appearing all of a sudden. These needs to explained in one line for clarity. Ex- phage therapy, CRISPR, Horizontal gene transfer, etc.

Line 313: How is prebiotics or probiotics related to epigenetic modulation in shrimp?

It will be nice to show a figure summarizing an overview of shrimp disease management startegies and how epigenetic regulations are connected with them or better than them.

4. The Concept of Epigenetics, Epigenetic Inheritance, and Its Potential Effect on Disease Management in Shrimp:-

Line 335: ‘Shrimp aquaculture, as a pivotal component of global food production, confronts a multitude of challenges, including the emergence of diseases that exert a substantial impact on shrimp health and production’. This similar statement is already mentioned above. Consider changing this statement.

Line 344: While epigenetics can play a role in the adaptation and response to environmental stimuli, it is important to acknowledge that it is only one of many factors influencing these processes. This statement means like epigenetics is the most important one. Epigenetic changes are not the sole determinants of an organism's ability to adapt to its environment or respond to external cues. Other genetic, environmental, and physiological factors also contribute significantly to the overall adaptability and responsiveness of an organism. Consider changing this statement and rewrite in context of both epigenetics and molecular mechnisms.

Line 351: What specific epigenetic marks or modifications are being considered for modification in shrimp to enhance disease resistance?

Line 355: Epigenetic modifications may not always lead to the desired outcomes, and there could be unintended consequences that need careful consideration. Does epigenetic modifications always enhance shrimps immune system?

Line 359: What is the concept of transgenerational epigenetic inheritance related to this review paper? Could the authors elaborate on the mechanisms proposed for transgenerational epigenetic inheritance in the context of the discussed review paper? It needs to be elaborated based on recent publications and how they can be linked with shrimp.

5.Types of Epigenetic Modifications:-

5.1. DNA Methylation:-

Line 375: There is a specific reason for extensive study of DNA methylation in the field of epigenetics. It will be nice to include them in the review paper.

Line 378: Which dnmts have been identified in shrimp? Write the name of dnmts discovered specific to shrimp and their function. Moreover, gene names should be italised in vertebrates.

Line 411: What is the role of non-cpg methylation in shrimp? Citing some references will be interesting for this line.

5.3. Elements of RNA

Line 566: Consider changing the heading to non-coding RNAs.

Line 573: Although RNAi is prevalent in many organisms, it is not universally conserved. Some eukaryotes, including certain fungi and protists, lack RNAi machinery or show significant variations in RNAi-related components. Consider changing the line accordingly.

Line 629: Piwi-like RNA have been studied in shrimp. It will be nice to include some examples from recent research works. Ex- https://doi.org/10.1186/s12864-022-08802-3

6. Applications of Epigenetics for Immunity Enhancement and Disease Management in Shrimp 684 Aquaculture

This section in well written with proper citations and discussion.

7. Challenges and Future Perspectives of Epigenetics in the Disease Management of Shrimp Aquaculture

Line 771-795: This section can be improved by addressing and citing references based on Limited Understanding of Shrimp Epigenome, Environmental Influences, Epigenetic Complexity, Transgenerational Epigenetic Effects, Validation of Epigenetic Targets and ethical considerations.

Line 796: The future perspective related to epigenetic modification is the most important part of this review paper. Authors should consider including the following oulines for this ection that can improve the credibility of the paper. It is important to write more about epigenetic profiling and comparative studies, transcriptomics-integrated epigenetics, epigenetic editing tools (like development of targeted DNA demethylation or histone modification systems), environmental management for epigenetics, and some epigenetic biomarkers for disease diagnosis in shrimp.

Line 882: Scientific name should be italised.

Overall, the review paper has the potential to make a significant contribution to the field of epigenetics in shrimp aquaculture and disease management. I encourage the authors to incorporate the suggested revisions and ensure that the scientific content is well-supported and up-to-date to enhance the credibility and impact of the paper.

Minor editing of English language required

Author Response

Dear reviewers,

We extend our heartfelt appreciation for dedicating your valuable time to thoroughly assess our manuscript titled "Epigenetic Modulations for Prevention of Infectious Diseases in Shrimp Aquaculture," and for offering your invaluable insights. Your constructive feedback has been immensely beneficial, and we have diligently addressed the highlighted areas in line with your guidance. To facilitate your review, the revised sections have been marked in green. We have incorporated the latest data to enhance the accuracy and relevance of the manuscript, and also refined certain statements to ensure clarity and precision.

Reviewer 2 Report

The review of Gunasekara CWR et al. discusses the role of Epigenetics in shrimp aquaculture that offers immense potential for disease control, productivity enhancement, and population robustness. Realizing epigenetic modulation's power requires overcoming complex challenges through innovative research, precise tools, monitoring, and integrating insights into breeding and aquaculture. Advances in feed, nutraceuticals, drugs, RNA interference, and CRISPR-Cas9 open ways to boost shrimp capabilities via epigenetic changes. However, the field is young, with many mechanistic mysteries to explore. Delving into epigenetic frontiers could establish a highly efficient shrimp farming industry, aligning prosperity with ecological stewardship.

The review represent a good significant in the field and could be interesting to the readers, however it need some improvement to be condensed to send a direct message to the readers and scientists.

Several sentences are difficult to read and there are several miss use of terms, the manuscript needs native proofreading, such as “indispensable, burgeoning, revolutionizing, accentuates, harnessing ……”. I understand the author needs to reduce the similarity but using non-common and may non-suitable words is not the way to overcome this issue.

The sections: 1 introduction and 2 Current Status of Global Production and Challenges in Shrimp Aquaculture could be reduced to half.

Line 89-93 is a very long sentence.

Fig 1 and 2. I did not recommend using FAO figures it could be deleted or modified/redesigned to be own to the author.

L401-423 about DNA methylation in vertebrates could be removed, to focus on the target.

In this section “5. Types of Epigenetic Modifications “could you focus on shrimp and crustacean and the role of epigenetics in diseases management rather than lecturing about the methods in several non-target animals.

Several sentences are difficult to read and there are several miss use of terms, the manuscript needs native proofreading

Author Response

Author response- Response to Reviewer 2

Dear reviewers,

We extend our heartfelt appreciation for dedicating your valuable time to thoroughly assess our manuscript titled "Epigenetic Modulations for Prevention of Infectious Diseases in Shrimp Aquaculture," and for offering your invaluable insights. Your constructive feedback has been immensely beneficial, and we have diligently addressed the highlighted areas in line with your guidance. To facilitate your review, the revised sections have been marked in blue. 

The review of Gunasekara CWR et al. discusses the role of Epigenetics in shrimp aquaculture that offers immense potential for disease control, productivity enhancement, and population robustness. Realizing epigenetic modulation's power requires overcoming complex challenges through innovative research, precise tools, monitoring, and integrating insights into breeding and aquaculture. Advances in feed, nutraceuticals, drugs, RNA interference, and CRISPR-Cas9 open ways to boost shrimp capabilities via epigenetic changes. However, the field is young, with many mechanistic mysteries to explore. Delving into epigenetic frontiers could establish a highly efficient shrimp farming industry, aligning prosperity with ecological stewardship.

Response to Reviewer 2

The review represents a good significance in the field and could be interesting to the readers, however, it needs some improvement to be condensed to send a direct message to the readers and scientists. Several sentences are difficult to read and there are several miss use of terms, the manuscript needs native proofreading. Such as “indispensable, burgeoning, revolutionizing, accentuates, harnessing ……”. I understand the author needs to reduce the similarity but using non-common and may non-suitable words is not the way to overcome this issue.

Thank you for the suggestion provided earlier. We have thoroughly revised the manuscript, addressing both grammar and scientific content in accordance with your guidance. The terms you mentioned have been successfully replaced, leading to an enhanced quality of the manuscript. Your assistance has been greatly appreciated in refining our work. We appreciate your response on the usage of scientific language within the manuscript. The replaced terms are marked in blue color.

Line 7- The term indispensable revised to “significance role”

Line 13- The term burgeoning was removed from the sentence and revised as “the field of epigenetics emerges”

Line 18- The term revolutionizing revised to “thereby contributing to”

Line 22- The term accentuates revised to “conveys”

Line 31- The term indispensable revised to “plays an important role”

Line 66-69-This sentence was revised by replacing the terms “burgeoning and harnessing”

Recently, there has been a growing interest in utilizing "targeted epigenetic modulations (purposefully changing the epigenome to obtain specific outcomes)" to enhance the biological capabilities of aquatic animals, including fish and shellfish [15,20]

Line 194-196- the term “burgeoning” was removed and the sentence was revised as follows

In recent times, the utilization of bacteriophages, vaccines, immune stimulants, and feed additives have emerged as viable alternatives to combat infectious diseases in the shrimp farming industry

Line 517-519- This sentence was revised as follows.

However, it is imperative to acknowledge that the exploration of histone modifications in crustaceans is a relatively new field of study, requiring additional scientific investigations to elucidate the intricate mechanisms that govern gene expression and cellular processes in these organisms.

Line 616- The term harnessing revised as “utilizing”

Line 680- The term indispensable revised as “essential”

Sections: 1 Introduction and 2 Current Status of Global Production and Challenges in Shrimp Aquaculture could be reduced to half.

We deeply appreciate your thorough review of this section.

Following your suggestion, we have diligently revised both Chapter One and Chapter Two, condensing their content (to 3 pages) while ensuring that the essential concepts remain effectively conveyed. Our aim was to enhance the clarity and coherence of the chapters, making them more reader-friendly and aligned with the overall flow of the manuscript.

Lines 89-93 is a very long sentence.

Thank you for the valuable suggestion on these lines. We have revised this sentence and separated it into two parts as follows.

It further underscores the importance of utilizing epigenetic regulatory mechanisms to ensure the long-term viability of the shrimp farming sector as a sustainable alternative to conventional disease control methodologies.

Additionally, this review will discuss the recent applications of epigenetic research, thereby presenting promising opportunities to develop targeted and precision-based interventions aimed at preventing infectious diseases in relation to shrimp aquaculture.

Fig 1 and 2. I did not recommend using FAO figures it could be deleted or modified/redesigned to be owned by the author.

The above mentioned figures namely, Figure 1 and Figure 2 are not FAO figures. They are original figures created using PPT by the authors, utilizing the statistics in the FAO for shrimp production. (Original figure files are attached in the supplementary files)

L401-423 about DNA methylation in vertebrates could be removed, to focus on the target, and in this section

We sincerely appreciate your thorough review of our work. In response to your comments, we'd like to address the key point you raised regarding Section 5.

L401-423: We greatly appreciate your insightful feedback concerning the DNA methylation segment and its relevance to the overarching objectives of our work. Your perspective highlights the importance of striking a delicate balance between content streamlining and presenting a comprehensive picture. In this part, we thoroughly believe that employing a comparative analysis of vertebrates and invertebrates has the potential to enrich the comprehension of distinct epigenetic mechanisms. By exploring the contrasts between these two broad categories of organisms, we not only deepen the understanding of the underlying principles but also set the stage for a more holistic grasp of the intricate dynamics of epigenetic mechanisms, particularly focusing on the context of our primary focus—shrimps. This comparative context serves as a valuable tool to elucidate the uniqueness of shrimp epigenetics, emphasizing the specific challenges and opportunities presented by these aquatic creatures. While our commitment to streamlining the content remains steadfast, we recognize the added value in retaining this comparative framework. Therefore, we intend to delicately integrate this perspective, ensuring that it complements and enhances the primary narrative centered on shrimp epigenetics (Therefore, we intend to keep this part as it is).

(Epigenetic mechanisms, which encompass heritable changes in gene expression not rooted in alterations to the DNA sequence itself, exhibit notable distinctions between vertebrates, animals with a backbone, and invertebrates, those lacking a backbone. Vertebrates engage extensively with DNA methylation, the addition of methyl groups to cytosine bases, and orchestrating gene silencing during vital processes such as development and differentiation. Contrarily, invertebrate DNA methylation is marked by variability and reduced prevalence)

  1. Types of Epigenetic Modifications “Could you focus on shrimp and crustaceans and the role of epigenetics in disease management rather than lecturing about the methods in several non-target animals?

We would like to express our sincere appreciation for your valuable suggestion to accentuate the focus on shrimp and crustaceans, particularly within the "Types of Epigenetic Modifications" section. Your input has prompted us to conduct a thorough review, carefully assessing the alignment of your suggestion with the overall scope and objectives of our manuscript. While we acknowledge the merit of narrowing the spotlight on shrimp and crustaceans in the aforementioned section, we have deliberated extensively and opted to maintain the current content intact. Our rationale behind this decision is rooted in our intention to provide a robust foundational understanding of epigenetic modifications in invertebrates, including the vital comparative perspectives with other animal groups (This approach is integral in establishing a comprehensive framework that illuminates the distinct epigenetic dynamics in shrimps, while simultaneously acknowledging the broader tapestry of epigenetics in the animal kingdom). {We have provided several examples (As in lines 466-473,482-492, 509-515,689-704), of how these mechanisms are involved in the prospect of shrimp}. We wish to underscore that your suggestion has undeniably enriched our considerations for both future research endeavors and the ongoing development of this manuscript. It has spurred us to reflect deeply on the balance between comprehensive exploration and focused analysis. We want to assure you that the subsequent sections of our manuscript, most notably Chapter 6, delve into an intricate discussion of the implications of "epigenetics in disease management and immunity enhancement." We are committed to ensuring that the richness of these later discussions effectively addresses the practical applications and outcomes, reflecting the very essence of your insightful recommendation.

We sincerely appreciate your engagement with our work and your commitment to enhancing its quality. Thank you. 

Reviewer 3 Report

the work Epigenetic Modulations For Prevention of Infectious Diseases in Shrimp Aquaculture presents a compendium of information concerning the prevention of pathologies in shrimp. In my opinion, it is a well-written and organized work, clear and of value to the scientific community. I present only 2 or 3 points that need to be revised, (see pdf).

Author Response

Author response- Response to Reviewer 3

Dear reviewers,

We express our gratitude for your valuable time spent on the thorough evaluation of our manuscript titled " Epigenetic Modulations for Prevention of Infectious Diseases in Shrimp Aquaculture " and for your insightful recommendations. We have diligently addressed the issues raised in accordance with your guidance, and the revised portions are duly highlighted in yellow for your convenience.

Response to Reviewer 3

The work Epigenetic Modulations for Prevention of Infectious Diseases in Shrimp Aquaculture presents a compendium of information concerning the prevention of pathologies in shrimp. In my opinion, it is a well-written and organized work, clear and of value to the scientific community. I present only 2 or 3 points that need to be revised.

Line 271- Something is missing. "Comma" "period" or text. please check.

Thank you for the response on the language used within the manuscript. Following your suggestions, the sentence was revised by removing the “comma” in the sentence line explored in shrimp aquaculture

structurally resembling cellular membranes, have been explored in shrimp aquaculture and notably,

Line 483- relying or it relies. A word seems to be missing, please check.

We acknowledge your observation regarding the missing word. It was missing the term “that”, and we have added and revised the sentence as follows.

Among different mechanisms used, histone acetylation is the most extensively studied histone modification that relies on the enzymatic actions of histone acetyltransferases (HATs) and histone deacetylases (HDACs)

Line 486- Consider removing this section to make it clearer “on the organisms”

We value your suggestion to remove the “on the organisms “part. following your suggestion, we have removed that to enhance the clarity of the sentence and revised it as follows.

Recent research in vertebrates and invertebrates has shown that histone acetylation primarily occurs on four histone proteins (H2A, 487 H2B, H3, and H4), with H3 and H4 exhibiting the highest levels of acetylation

Figure 5: The image needs to be revised as follows

We value your suggestion changing the Figure 5 terms of SiRNA, MiRNA, Gene, PiRNA, and LncRNA which contained capital letters within the image. It has been revised as suggested.

SiRNA-siRNA, MiRNA-miRNA, Gene-gene, PiRNA-piRNA, LncRNA-lncRNA

Line 822- should be Italic and Magallana gigas is the currently accepted name.

We value your suggestion on the scientific name and replacing it with the currently accepted name.

study of Pacific oysters (Magallana gigas) may also offer

We sincerely appreciate your engagement with our work and your commitment to enhancing its quality. Thank you.

Round 2

Reviewer 1 Report

The authors improved the manuscript. My recommendation is accept the manuscript in the present form

Minor editing of English language required

Reviewer 2 Report

The manuscript was improved substantially and could be accepted in its current form.